# Collagen Lattice Model, Populated with Heterogeneous Cancer-Associated Fibroblasts, Facilitates Advanced Reconstruction of Pancreatic Cancer Microenvironment

**DOI:** 10.3390/ijms25073740

**Published:** 2024-03-27

**Authors:** Xiaoyu Song, Yuma Nihashi, Yukiko Imai, Nobuhito Mori, Noritaka Kagaya, Hikaru Suenaga, Kazuo Shin-ya, Masamichi Yamamoto, Daiki Setoyama, Yuya Kunisaki, Yasuyuki S. Kida

**Affiliations:** 1Tsukuba Life Science Innovation Program (T-LSI), School of Comprehensive Human Sciences, University of Tsukuba, Tsukuba 305-8572, Japan; work.xiaoyu93@gmail.com; 2Cellular and Molecular Biotechnology Research Institute, National Institute of Advanced Industrial Science and Technology (AIST), Tsukuba 305-8565, Japan; y-nihashi@aist.go.jp (Y.N.); n-mori@aist.go.jp (N.M.); noritaka.kagaya_n2pc@natprodchem.jp (N.K.); suenaga-hikaru@aist.go.jp (H.S.); k-shinya@aist.go.jp (K.S.-y.); 3Department of Plastic and Reconstructive Surgery, Faculty of Medicine, University of Tsukuba, Tsukuba 305-8575, Japan; tkb101y.ima@gmail.com; 4Department of Research Promotion and Management, National Cerebral and Cardiovascular Center, Ki-shibe-Shimmachi, Suita 564-8565, Japan; myamamoto@ncvc.go.jp; 5Department of Clinical Chemistry and Laboratory Medicine, Kyushu University Hospital, Fukuoka 812-8582, Japan; setoyama.daiki.753@m.kyushu-u.ac.jp; 6Department of Clinical Chemistry and Laboratory Medicine, Graduate School of Medical Sciences, Kyushu University, Fukuoka 812-8582, Japan; kunisaki.yuya.519@m.kyushu-u.ac.jp; 7School of Integrative & Global Majors, University of Tsukuba, Tsukuba 305-8572, Japan

**Keywords:** pancreatic ductal adenocarcinoma, tumor microenvironment, cancer-associated fibroblasts, 3D tumor model, drug screening

## Abstract

Pancreatic ductal adenocarcinoma (PDAC) is a solid-tumor malignancy. To enhance the treatment landscape of PDAC, a 3D model optimized for rigorous drug screening is essential. Within the PDAC tumor microenvironment, a dense stroma comprising a large extracellular matrix and cancer-associated fibroblasts (CAFs) is well-known for its vital role in modulating tumor growth, cellular heterogeneity, bidirectional paracrine signaling, and chemoresistance. In this study, we employed a fibroblast-populated collagen lattice (FPCL) modeling approach that has the ability to replicate fibroblast contractility in the collagenous matrix to build dense stroma. This FPCL model allows CAF differentiation by facilitating multifaceted cell–cell interactions between cancer cells and CAFs, with the differentiation further influenced by mechanical forces and hypoxia carried within the 3D structure. Our FPCL models displayed hallmark features, including ductal gland structures and differentiated CAFs with spindle shapes. Through morphological explorations alongside in-depth transcriptomic and metabolomic profiling, we identified substantial molecular shifts from the nascent to mature model stages and potential metabolic biomarkers, such as proline. The initial pharmacological assays highlighted the effectiveness of our FPCL model in screening for improved therapeutic strategies. In conclusion, our PDAC modeling platform mirrors complex tumor microenvironmental dynamics and offers an unparalleled perspective for therapeutic exploration.

## 1. Introduction

Pancreatic cancer is a persistent malignancy with one of the lowest 5- and 10-year survival rates compared to other tumor types [1] and is projected to become the second-leading type of cancer death by 2030 [2]. Pancreatic cancer is asymptomatic in its initial stages and is located deep in the abdominal cavity, making it challenging to diagnose [3]. Most patients with pancreatic cancer are diagnosed at advanced stages, wherein the disease progresses swiftly and efficacious treatments remain limited [4,5,6].

Pancreatic ductal adenocarcinoma (PDAC) is the predominant and highly aggressive form of pancreatic cancer, accounting for >85% of all diagnoses [7]. Characterized by its rich extracellular matrix (ECM) and complex cellular constituents, PDAC contains a substantial proportion of heterogeneous cancer-associated fibroblasts (CAFs), including myofibroblastic CAFs (myCAFs), inflammatory CAFs (iCAFs), and antigen-presenting CAFs (apCAFs) [8,9,10]. Such heterogeneous fibroblasts form a dense fibrotic stroma, which not only supports tumor growth but also contributes to chemoresistance through mechanisms such as altering cancer cell signaling [11,12,13,14,15]. Understanding the complex interactions within the tumor microenvironment is crucial for developing strategies to overcome drug resistance and improve treatment outcomes. Additionally, this dense fibrotic stroma, along with the ECM, acts as a barrier around the tumor to impede drug penetration [16,17] and mitigate the therapeutic effects of targeted therapies and immunotherapies [18,19], which have demonstrated promise in other malignancies, such as melanoma [20], lung cancer [21], and breast cancer [22]. Given the lack of breakthroughs in PDAC treatment, a reliable scanning model for potential drug candidates is urgently required. To simulate the PDAC microenvironment, we targeted the replication of heterogeneous CAFs. Adipose-derived mesenchymal stem cells (AD-MSCs), renowned for their expansive multilineage potential and self-renewal capacity, have been used as CAF progenitors [23,24,25,26]. In our previous studies, we demonstrated that CAF progenitors co-cultured with PDAC cells could differentiate into heterogeneous CAFs both in vitro [23] and in vivo [27], forming a PDAC-like clinical morphology with ductal gland epithelial cancer cells surrounded by abundant stromal cells.

Conventional 2D monolayer culture does not adequately capture the multifaceted spatial cell signaling and associated mechanical stress inherent in the ECM [28,29,30]. Furthermore, these 2D models cannot replicate the nutrient-deprived and hypoxic conditions of PDAC. Reproducing these properties within the model can restore drug sensitivity and resistance in PDAC, thereby improving the precision of drug efficacy prediction [31,32]. Although in vivo PDAC models can overcome these drawbacks, their protracted development timelines, fiscal implications, and ethical constraints impede pharmacological exploration [33]. Therefore, following the previous strategy for heterogeneous CAFs, we attempted to construct a 3D PDAC model that could build a bridge between 2D in vitro and in vivo models [34]. Our novel 3D PDAC model aimed to replicate the high density and stiffness characteristics of PDAC. To achieve this, we employed the fibroblast-populated collagen lattice (FPCL) model, a classical wound repair model developed by Bell et al. [35]. This model relies on the contraction force exerted by fibroblasts in the collagen gel, which eventually forms a disc-like structure. FPCL contraction dynamics may have the potential to spatially reproduce mechanical stress in the PDAC microenvironment. The high expression of mechanical stress is a unique feature in PDAC, playing a substantial role in reproducing the PDAC microenvironment [36]. Our 3D PDAC tumor model emphasizes the unique strength of physically replicating the tumor microenvironment. This sets it apart from commonly used 3D models that exclusively focus on biological replication, including models that incorporate patient-derived PDAC cells and CAFs [37].

In this study, we developed a novel 3D PDAC model exhibiting clinically similar glandular structures surrounded by spindle-shaped stromal cells. This model could replicate certain biological and physical properties, as evidenced by the analysis of the global transcriptome. Metabolomic analysis revealed potential biomarkers associated with the hypoxic tumor microenvironment that could provide valuable insights for PDAC diagnosis. Additionally, our PDAC-FPCL model demonstrated high tumorigenic potential in vivo. Compound screening trials have indicated that a combined therapeutic approach using anticancer and antifibrotic agents results in a better prognosis. These findings underscore the importance of targeting stromal cells in PDAC for therapeutic development.

## 2. Results

### 2.1. Morphological and Contractile Dynamics of PDAC-FPCL Models

Using a collagen matrix supplemented with both PDAC and CAF progenitor cells, we established a combined PDAC-FPCL model (referred to as the comPDAC-FPCL model), wherein PDAC cells (Capan-1) and CAFs coexist (Figure 1A). Adapting to various experimental scales, both 24- and 96-well plates were harnessed, facilitating the derivation of the comPDAC-FPCL models across different dimensions, as shown in Figure 1B,C, respectively. In addition, 384-well plates were utilized (Appendix A). In the matrix without cellular constituents (designated as Collagen-only), discernible contractile activity was absent. Although the Collagen-Capan-1 group exhibited a certain degree of contractility, attaining the level observed in the comPDAC-FCPL model proved challenging. This discrepancy indicates a difficulty in accurately representing the stiffness characteristics of PDAC. The contractile trajectory of the comPDAC-FPCL models aligned with that of the CAF-FPCL group, indicating that the primary contractile impetus was derived from CAF progenitors. Notably, comPDAC-FPCL reduced the model size between Days 7 and 9. Hematoxylin and eosin (H&E) staining was used for the in-depth morphological assessment of comPDAC-FPCL models. This analytical approach identified the presence of mature ductal glands, reminiscent of clinical observations, in both 24- and 96-well plates on Days 7 and 9, respectively (Figure 1D,E).

### 2.2. Differential Morphological Dynamics of CAFs within PDAC-FPCL Models

The exploration of morphological dynamics is crucial for understanding the differentiation trajectory of CAFs. Capan-1 cells were identified using the epithelial cell adhesion molecule (EpCAM), a hallmark marker of epithelial malignancies. CAFs were characterized using vimentin, a recognized fibroblast marker. The glandular structures marked by EpCAM within the comPDAC-FPCL matrix manifested a complex and irregular expansion trend (Figure 2A). By Day 7, the integrated CAFs adopted a spindle-shaped morphology, which aligned with the typical CAF morphological characteristics observed in clinical PDAC microenvironments [27]. These spindle-shaped CAFs were in stark contrast to the round-shaped CAF progenitors present in the CAF-FPCL model. These morphological features were also confirmed in the 3D reconstruction images (Figure 2B, Appendix A), revealing 3D glandular structure formations by PDAC cells and CAF-surrounded glands. To further explore CAF morphological transitions (Figure 2C), assessments were performed at predetermined culture intervals. The salient transition of CAFs into a stellate spindle shape was apparent between Days 7 and 9. This trend is consistent with the calculated Shape Index (Major Axis/Minor Axis ratio), which peaked during this period (Figure 2D). Furthermore, vimentin-positive cell quantification determined an apex in the mean cellular area by Day 9 (Figure 2E). The number of Hoechst-positive cells remained consistent. Collectively, these findings highlight the dynamic morphological evolution of CAFs in the comPDAC-FPCL model.

### 2.3. Differential Expression and Temporal Dynamics of CAF Subtypes

To investigate the differentiation heterogeneity of CAFs in the comPDAC-FPCL models, myCAF and iCAF markers were examined by quantitative PCR on Days 0, 8, and 14 (Figure 3A). The expression profiles revealed a surge in myCAF markers on Days 8 and 14, such as *actin alpha 2* (*ACTA2*) and *Tropomyosin* (*TPM1*). Conversely, Leukemia Inhibitory Factor (*LIF*), identified as one of the relevant markers for iCAFs, exhibited elevated expression in the subsequent stages of comPDAC-FPCL formation, which may imply that the hypothesis of iCAFs differentiation within the model is sustained. The dynamics of gene expression are shown in Appendix A. Nevertheless, further experiments employing a broader range of markers are required to substantiate this speculation. In a complementary assessment, the secretion of connective tissue growth factor (CTGF), an indicator of mechanical stress, was quantified to gauge the physical and mechanical attributes of the FPCL models. Given that CTGF expression is intricately linked to mechanical stress responses, as evidenced by previous studies [38,39], it was imperative to examine both the CAF-FPCL and comPDAC-FPCL models on Days 7 and 14. The findings revealed that CTGF levels were markedly elevated in contrast to those in the 2D culture milieu, and no statistically significant divergence was observed between the CAF-exclusive and combined models (Figure 3B).

### 2.4. Transcriptomic Dynamics Show Structural and Metabolic Alterations

To understand the shifts in the global transcriptomic landscape during maturation of the comPDAC-FPCL models, RNA sequencing (RNA-seq) of the models was performed on Days 0, 7, and 14, followed by a Gene Ontology (GO) analysis. Remarkably, Days 7 and 14 highlighted a significant enrichment in biological processes integral to the active tumor microenvironment. These processes spanned areas such as “cell adhesion” and “ECM organization” and included the emergence of “inflammatory response” (Figure 4A,B, Appendix A). To further determine the temporal alterations that occurred between Days 7 and 14, a side-by-side comparison of gene expression was performed (Figure 4C,D, Appendix A). Notably, “inflammatory response” was exhibited in the comPDAC-FPCL model on Days 7 and 14, yet with distinct gene compositions. This implies that the comPDAC-FPCL model displayed varying complexity and biological resemblance on Days 7 and 14, resulting in different profiles of inflammatory-related genes. On Day 7, the comPDAC-FPCL models distinctly emphasized a synchronized tumor microenvironment. This was shown by the upregulated pathways, including “inflammatory response”, “cell adhesion”, “ECM organization”, and “collagen fibril organization”. In contrast, Day 14 presented a surge in metabolic pathways, incorporating processes like “glucose metabolic process”, “retinoic acid metabolic process”, and “cellular response to fatty acid”. Similar to these findings, the volcano plot analysis highlighted genes pivotal to the ECM. They are also known as CAF markers, namely *POSTN*, *COL1A1*, *CXCL12*, *VIM*, *COL1A2*, and *LRRO15*, and were predominantly expressed in the Day 7 configurations.

### 2.5. Characterization of Extracellular Metabolite Dynamics

Our RNA-seq analysis indicated marked alterations in the metabolic processes within the comPDAC-FPCL models, aligning with our prior data. This highlighted the association between tumor microenvironment maturation and metabolic pathways, particularly pertaining to extracellular metabolite dynamics. Thus, comprehensive metabolic profiling is imperative [40,41]. We subjected the extracellular primary metabolites from the harvested supernatants to rigorous liquid chromatography–mass spectrometry. The resultant metabolomic landscape is portrayed in a heatmap (Figure 5A), with special emphasis on the top 25 metabolites with significant differential abundance. The supernatants derived from the Collagen-only models were observed to replete with media-derived amino acids, including pivotal amino acids such as leucine, lysine, and tryptophan. In contrast, supernatants from the Collagen-Capan-1 models showed augmented levels of glutamic and lactic acids. In contrast, the CAF samples predominantly contained isocitric and citric acids, echoing the characteristic metabolic signatures of PDAC cells and CAF progenitors. A salient feature was the discernible enrichment of 4-hydroxyproline and proline in comPDAC-FPCL models. Recognizing these metabolites as proposed biomarkers for PDAC differentiation [42], our findings substantiate the capability of comPDAC-FPCL models to recapitulate the metabolic characteristics of the PDAC microenvironment. The indispensable role of proline, particularly in facilitating cancer cell proliferation during hypoxia, highlights its significance. In this context, to validate the hypoxic conditions presented by the comPDAC-FPCL models, we evaluated the expression levels of quintessential hypoxia markers, *HIF1A* and *EPAS1* (Figure 5B). The increase in their expression by Day 14 was consistent with our postulation that the models adeptly mirror the hypoxic environment and concomitant metabolic reconfigurations typical of advanced PDAC.

### 2.6. Validation of Tumorigenic Potential of 3D Tissues Generated by the comPDAC-FPCL Models In Vivo

A comprehensive in vivo study was conducted to elucidate the oncogenic potential of CAFs in 3D FPCL models. Specifically, the Collagen-Capan-1 model and its counterpart, comPDAC-FPCL, were subcutaneously transplanted into immunodeficient murine hosts (Figure 6A). As anticipated, the tumor volumes from the comPDAC-FPCL transplants surpassed those from the Collagen-Capan-1 group (Figure 6B), underscoring the pivotal role of CAFs in accelerating tumor growth. For a more comprehensive understanding, H&E staining was used to examine the histological architecture of the tumors. This histochemical investigation revealed a marked stromal signature in comPDAC-FPCL sample tumors (Figure 6C). Immunohistochemical evaluations were performed to discern the cellular dynamics within the tumor microenvironment. The Collagen-Capan-1 transplant tumors predominantly exhibited MUC1-positive ductal cancer formation, interspersed with occasional vimentin- or α-SMA-positive fibroblasts. In contrast, comPDAC-FPCL-derived tumors were characterized by the dense presence of vimentin- and α-SMA-positive fibroblasts, emphasizing their stroma-rich nature (Figure 6D). This evaluation underscores the pronounced oncogenic capabilities of the comPDAC-FPCL models.

### 2.7. Applications of comPDAC-FPCL Models in Drug Screening

In the field of targeted therapeutics, the ability to screen drug efficacy in representative models is crucial. With this intent, we developed a consistent protocol for screening potential drug candidates using comPDAC-FPCL models (Figure 7A). This regimen spanned six days, with interim medium replacement on Day 6. On the 9th day, the impact on the comPDAC-FPCL models was assessed to identify promising drug candidates. Gemcitabine (Gem) was selected as a preliminary test because of its known anticancer properties. Remarkably, the comPDAC-FPCL models exhibited a considerable increase in size when exposed to a higher Gem concentration of 50 μM (Figure 7B,C). This size modulation indicated its potential as a metric for assessing drug efficiency, and simultaneously, it might indicate that high-dose Gem causes changes in CAF activity, including the loss of collagen gel contraction ability. We posit that the structural integrity of the FPCL models is contingent on cellular contractile forces. If these cells undergo damage, such as cell death, their integrity may be compromised. To validate this, a Cell Counting Kit-8 (CCK-8) assay was used to assess cell viability, with findings echoing the size-based observations (Figure 7D). Expectedly, we observed differential therapeutic responses to the applied drug concentrations of 50, 10, and 2 μM. Initially, we reported that treatments at 50 μM concentration showed pronounced therapeutic advantages, characterized by significant reductions in PDAC cell viability and notable disruptions to the collagen matrix integrity. At lower concentrations of 10 and 2 μM, while the therapeutic outcomes were less dramatic, they were still evident. Subtle yet significant changes in cell viability and matrix structure at these concentrations suggest a dose-dependent effect. This observation highlights the nuanced nature of drug response in our PDAC model, underscoring the fact that even at lower concentrations, there is a detectable impact on tumor cells and their microenvironment, albeit less pronounced. To provide a clearer understanding of these effects in the comPDAC-FPCL model, we have included a more in-depth analysis of 3D reconstructed imaging in Figure 7E (Appendix A), detailing how each concentration impacts PDAC cell behavior and the structural integrity of the collagen matrix. PDAC cells exhibited a response to Gem treatment, leading to the disruption of glandular structures in all treatment groups. Moreover, the survival of CAFs was challenging at a high concentration of 50 μM Gem, making maintenance of the contractile structure of comPDAC-FPCL models difficult. To broaden the study scope, a multidrug combinatorial approach was explored. When Gem and the anticancer drug Paclitaxel (Pac) were applied to the model, high-dose Pac was slightly effective. However, when Gem was paired with the idiopathic pulmonary fibrosis drug Pirfenidone (Pir) as a stromal modulator, the combined therapeutic effect was markedly superior to their individual effects. Notably, increasing the dose of Pir further amplified this therapeutic outcome (Figure 7F,G). In essence, these findings not only indicate the robustness of the comPDAC-FPCL models in drug screening but also emphasize their potential in tailoring multi-drug treatment strategies for PDAC interventions.

## 3. Discussion

We verified that the interior of the comPDAC-FPCL model possessed a morphology similar to that of previous in vitro and in vivo models, characterized by spindle-shaped CAFs surrounding the pancreatic glandular structures. This implies that the strategy of CAF maturation and differentiation through interaction with PDAC cells was also applied to 3D collagen lattice models, as demonstrated by the myCAF and iCAF differentiation in the comPDAC-FPCL model. However, the absence of immune cells in these models hinders apCAF differentiation, which is typically influenced by macrophages and dendritic cells [43]. Hence, owing to the lack of PDAC components, the complete replication of PDAC heterogeneity remains unattainable at this stage. To replicate the intricate tumor microenvironment of PDAC, future studies should incorporate the missing endothelial cells, immune cells, and various other stromal components into the construction of comPDAC-FPCL models.

PDAC stands out among solid human carcinomas because of its exceptional stiffness and distinct desmoplastic reactions that are closely associated with mechanical stress. This highly fibrotic tumor microenvironment triggers the malignant phenotype of PDAC, promoting cancer cell survival and migration [44]. In this study, CTGF, an indicator of mechanical stress, was identified in the conditioned medium of the comPDAC-FPCL model. To further characterize the density structure of the comPDAC-FPCL model, YAP/TAZ, according to the cellular microenvironment, was considered the sensor and mediator of mechanical cues in subsequent immunostaining experiments [45]. In addition, direct determination of the presence of mechanical stress and stiffness within the model is imperative. Young’s modulus, quantified using atomic force microscopy, has emerged as a valuable method for detecting the stiffness and elasticity of 3D models [46]. Mechanical stress can potentially induce interstitial fluid pressure, impede angiogenesis, and create a hypoxic and nutrient-deprived PDAC tumor microenvironment.

The elevated expression of proline and 4-hydroxyproline demonstrates the nutrient-deprived condition in the comPDAC-FPCL model, highlighting the critical role in maintaining PDAC growth through proline dehydrogenase-catalyzed TCA cycle metabolism for ATP production. This underlines the reasoning behind the recent focus on PRODH-targeted therapy using inhibitors in oncology, demonstrating the treatment efficacy of this strategy in both lung [47] and breast cancer [48]. Furthermore, it elaborates on the potential of proline as an indicative biomarker for diagnosing patients with PDAC [42]. Hypoxia exerts a crucial role on the inflammatory phenotype of PDAC, with its associated inducible factors providing essential conditions for the differentiation of iCAFs, including IL1α and IL6 [49,50]. Notably, the preferential localization of iCAFs within hypoxic regions of PDAC supports this characterization. Previous studies have demonstrated that HIF1α stabilization was adequate to induce an iCAF phenotype and enhance PDAC tumor growth [51], offering a theoretical foundation for the potential differentiation of iCAFs within our comPDAC-FPCL model. Consequently, hypoxia in PDAC serves as a potent regulator of CAF heterogeneity and a promoter of tumor progression in PDAC.

Collagen I, extensively produced by CAFs, plays a critical role in modulating tumor growth and progression. This dimer form of collagen not only facilitates a scaffold for tumor cell adhesion and migration, contributing to the invasive properties of PDAC, but also potentially impacts the delivery and efficacy of chemotherapeutic agents through the modification of the extracellular matrix’s mechanical properties. Studies by Chen et al. (2022) [52] and Tian et al. (2021) [53] have shed light on how collagen produced by CAFs can alter the mechanical properties of the extracellular matrix, consequently influencing tumor cell behavior and drug sensitivity. Building upon these findings, our study (Song, Bioengineering 2023 [54]) elucidated the role of collagen in the structural and functional maturation of tumors. Consequently, targeting collagen I dimers could offer a novel approach to disrupting the PDAC microenvironment, thereby enhancing therapeutic outcomes. This understanding underscores the importance of collagen’s presence as modulated by CAFs, revealing its significant influence on the heterogeneity of tumor growth patterns and the variable response to treatment in PDAC. Furthermore, the use of collagen is a substantial improvement over the lengthy and costly mouse model, supporting a prerequisite for large-scale drug screening. While type I collagen requires intricate processing, resulting in a relatively high cost, its expense has been greatly reduced compared to in vivo models. However, its implementation for extensive large-scale drug screening remains challenging. Consequently, economical synthetic materials will be considered in future research. Moreover, owing to rapid PDAC progression, drug screening models with shorter modeling times must be developed.

In preliminary drug-screening experiments, the comPDAC-FPCL models could identify effective drug candidates. In this case, combination treatment with anticancer and antifibrotic drugs (Gem+Pir) exhibited considerable efficacy against PDAC, highlighting the therapeutic potential of targeting CAFs. In addition, the combination therapy targeting only cancer cells (Gem+Pac) did not show additive therapeutic effects. One possible reason for this is that CAFs boost PDAC drug resistance by releasing extracellular vesicles, which can reduce the effective drug concentration within the tumor cells. Another possible reason is that the stiffness of the comPDAC-FPCL models may serve as an additional cause of paclitaxel resistance. PDAC stiffness has been shown to be linked to paclitaxel resistance, while not affecting resistance to gemcitabine [44]. Through this detailed examination, we aim to convey the complexity of therapeutic response in PDAC, demonstrating that while higher drug concentrations yield more immediate and observable effects, lower concentrations still possess therapeutic value, influencing tumor dynamics in a more subtle manner. This serves to better articulate the differential impact of drug concentrations on PDAC treatment outcomes, offering valuable insights for optimizing therapeutic strategies in combating this aggressive cancer. Collectively, the comPDAC-FPCL model holds promise as an experimental platform to simulate the PDAC microenvironment for drug screening.

Our comPDAC-FPCL models embody the unique attributes of PDAC, replicating its heterogeneity and pronounced stiffness, which correspond to the clinical morphologies of PDAC. These models mirror the hypoxic conditions and mechanical stress inherent in the PDAC microenvironment. This fidelity increases the accuracy of drug candidate screening. The efficiency of model establishment, coupled with its reproducibility, facilitates expansive drug screening. Additionally, it is imperative to emphasize the relevance of our model for both preclinical and clinical drug discovery efforts. The model’s ability to replicate the complex tumor microenvironment offers a unique platform for testing the efficacy of new therapeutic agents and combinations therapies. By providing insights into the interaction between cancer cells and the tumor microenvironment, our model paves the way for the identification of potential targets and the development of more effective treatments. Therefore, our model not only aids in enhancing our understanding of PDAC biology but also holds promise for accelerating the translation of research findings into clinical applications, ultimately improving patient outcomes.

In future perspectives, we aim to extend our 3D PDAC model into a more dynamic 4D model that incorporates time as a fourth dimension, enabling us to better mimic the evolving nature of the tumor microenvironment over time. This advancement will allow for the observation of drug responses and tumor progression in a manner that closely replicates in vivo conditions, offering a more accurate and comprehensive tool for preclinical drug testing and understanding the mechanisms of disease progression and resistance.

## 4. Materials and Methods

### 4.1. Cell and Culture Conditions

The immortalized human AD-MSC cell line ASC52telo (ATCC SCRC-4000) and human pancreatic cancer cell line Capan-1 (ATCC HTB-79) were used in this study and cultured following a previous protocol [23,27].

### 4.2. Construction of 3D PDAC-FPCL Models

To establish 3D PDAC-FPCL models, Capan-1 cells and AD-MSCs were suspended in a neutralized type I collagen solution (IAC-30; Koken Co., Tokyo, Japan) in a 1:2 ratio. Their densities were 1 × 10^5^/mL and 2 × 10^5^/mL, respectively. The Collagen-Capan-1 group was constructed using a type I collagen solution containing Capan-1 cells at a density of 1 × 10^5^/mL. Tissue culture plates were pre-coated with 2-methacryloyloxyethyl phosphorylcholine polymer (LIPIDURE-CM5206; NOF Co., Tokyo, Japan). In 24- and 96-well plates, 800 and 200 μL/well of the mixed collagen solution was added, respectively. The plates were incubated at 37 °C for 30 min for gelation, and the gels were released from their wells using a vortex mixer (GENIE 2; Scientific Industries Inc., Bohemia, NY, USA). The plates were incubated on a shaker at 95 rpm for extended culture. Fresh medium was substituted every 2 or 3 days. The area of the collagen lattice model was measured using ImageJ software (1.53c, Java 1.8.0_172; U.S. National Institutes of Health, Bethesda, MD, USA). The relative area of the collagen lattice model was calculated as the percentage of the lattice area to the initial gel area.

### 4.3. RNA Extraction and Quantitative Real-Time PCR (qPCR)

The methods of RNA extraction from cells and collagen discs, and cDNA conversion, were based on a previous study [55]. Briefly, total RNA was extracted from the cells using NucleoSpin RNA (Macherey Nagel GmbH & Co., KG, Duren, Germany) and from the collagen discs using ISOGEN reagent (Nippon Gene, Tokyo, Japan), following the manufacturer’s instructions. Subsequently, 1 μg of total extracted RNA was converted into cDNA using ReverTra Ace reverse-transcription reagents (TOYOBO, Osaka, Japan), following the manufacturer’s instructions. qPCR was performed using gene-specific PrimeTime qPCR probes (Integrated DNA Technologies, Coralville, CA, USA) and Thunderbird SYBR qPCR mix (TOYOBO), according to the manufacturer’s instructions. The expression levels of the target genes were normalized to those of vimentin. The primer sequences are listed in Appendix A.

### 4.4. RNA-Seq Analysis

RNA was isolated from comPDAC-FPCL models on Days 0, 7, and 14 using the ISOGEN reagent (Nippon Gene), following the manufacturer’s instructions. RNA-seq was performed as previously described [55]. GO enrichment analysis elucidated the functions of the differentially expressed genes using the Database for Annotation, Visualization, and Integrated Discovery (DAVID). A volcano plot was constructed using VolcaNoseR (https://huygens.science.uva.nl/VolcaNoseR/ (accessed on 11 April 2023)). The raw sequences in FASTQ format are available from DDBJ (DRA016977).

### 4.5. Conditioned Medium Collection and CTGF Analysis

The conditioned medium from PDAC-FPCL was collected on Days 7 and 14. All conditioned media were freshly replaced 72 h prior to collection; they were then centrifuged at 200× *g* for 10 min at 4 °C, followed by freezing and storage at −80 °C. To detect CTGF secretion, an enzyme-linked immunosorbent assay (Human CTGF ELISA Kit, ab261851; Abcam, Cambridge, UK) was performed according to the manufacturer’s instructions.

### 4.6. Histological Analysis

The paraffin-embedded sections were prepared for morphological and immunohistochemical analyses. comPDAC-FPCL was encapsulated in iPGell (Genostaff, Tokyo, Japan) before sectioning to preserve the structure. H&E and immunofluorescence staining were performed as previously described [56]. Subsequently, the primary antibodies against vimentin (1:1000, MAB2105; R&D system, Minneapolis, MN, USA), EpCAM (1:400, ab71916; Abcam), α-SMA (1:10,000, ab7817; Abcam), and MUC-1 (1:200, MA1-06503; Thermo Fisher Scientific, Waltham, MA, USA) were applied, and the corresponding secondary antibodies (1:200; Alexa Fluor 488-conjugated anti-mouse/rabbit, Alexa Fluor 555-conjugated anti-mouse/rabbit, Alexa Fluor 555-conjugated anti-rat; Thermo Fisher Scientific) were used. Additionally, Hoechst 33342 (1:500; Dojindo, Kumamoto, Japan) was used to counterstain the cell nuclei.

### 4.7. 3D Reconstruction Imaging of the comPDAC-FPCL Model

For 3D reconstruction imaging, the aforementioned 96-well plate comPDAC-FPCL models were employed. The models were sequentially submersed in CUBIC-L and CUBIC-R (T3740 and T3741, respectively; Tokyo Chemical Industry CO., LTD., Tokyo, Japan), according to the manufacturer’s instructions, for staining pretreatment. Thereafter, the models were stained with vimentin (1:200) and EpCAM (1:200) antibodies, followed by the corresponding secondary antibodies. Finally, the models were transferred to glass-bottom culture dishes and infiltrated using a mounting solution (RI 1.520). The 3D reconstruction images were captured using a Zeiss confocal microscope (LSM 700; Oberkochen, Germany).

### 4.8. Mouse Model and In Vivo Experiments

Seven-week-old female nude mice (BALB/c-nu/nu; CLEA Japan, Tokyo, Japan) were housed under specific pathogen-free conditions at the Animal Center of the National Institute of Advanced Industrial Science and Technology (AIST) and randomly allocated to two groups: a PDAC cell group (n = 3) and a comPDAC-FPCL group (n = 3). All FPCL models were prepared in 24-well plates and collected on Day 7. Thereafter, each set with three models was mixed with 100 μL Matrigel and solidified into tumor fragments at room temperature. These tumor fragments were then placed in 1 cm subcutaneous pockets on the left thigh. Tumor volume was measured biweekly post-transplantation. On Day 49, the mice were euthanized by cervical dislocation, and all subcutaneous tumors were excised. All invasive procedures were performed under isoflurane anesthesia. Notably, all aspects of the animal experiments and procedures adhered to the approved guidelines and received approval from the Institutional Animal Care and Use Committee of the respective institutes of AIST.

### 4.9. CCK-8 Assays

After drug treatment of the comPDAC-FPCL models, a CCK-8 assay (Dojindo) was used to detect cell viability within the models, following the manufacturer’s instructions. After adding the detection solution, the plates were incubated at 37 °C for 1 h. Next, 100 μL reaction solution from each well was transferred into 96-well plates. The absorbance was measured at 450 nm using a microplate reader (Thermo Fisher Scientific). Six replicate experiments were performed for each group, and the highest and lowest values were subtracted from the calculations.

### 4.10. Metabolomic Analysis

Primary metabolites were analyzed using liquid chromatography-mass spectrometry (LC-MS-8060, Shimadzu, Kyoto, Japan). Specific experimental methods were based on a previous study [41].

### 4.11. Statistical Analysis

Data are presented as mean ± standard deviation (SD). The significance of differences between the two groups was assessed using the Student’s *t*-test. Metabolite data were exported in CSV format and uploaded onto the MetaboAnalyst platform (accessible at [https://www.metaboanalyst.ca] accessed on 4 August 2023), which can comprehensively process and analyze metabolic profiles. A default data integrity check verified the data integrity, and filtering was performed based on the mean intensity value. *p* < 0.05 was considered to be statistically significant. Error bars represent SD.

## Figures and Tables

**Figure 1 ijms-25-03740-f001:**
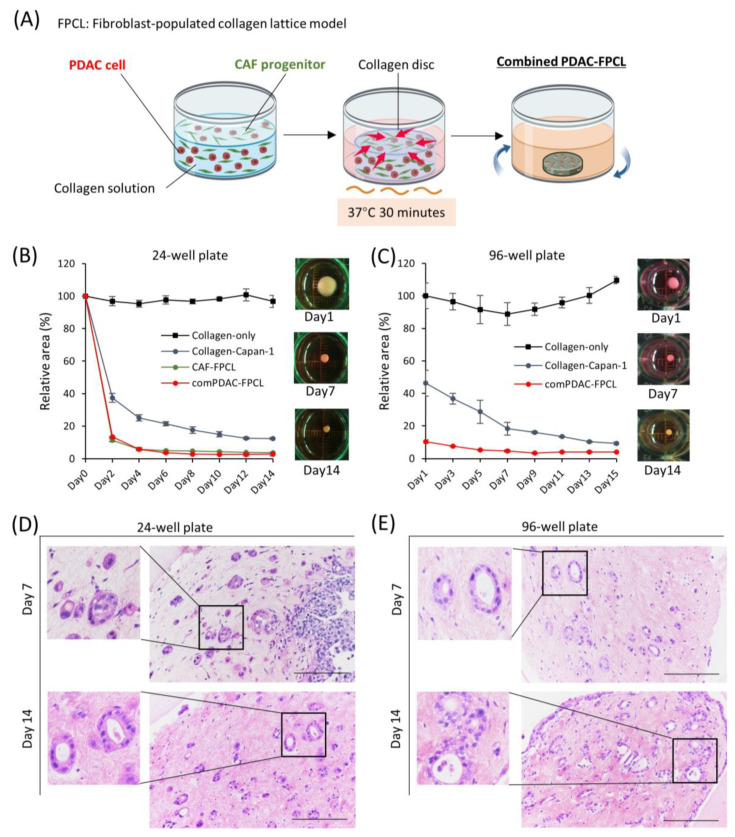
Morphological and contractile dynamics of comPDAC-FPCL models. (**A**) Graphical representation of the establishment of comPDAC-FPCL models. The rotating blue arrows represent the incubation in a shaker. The contraction formation of comPDAC-FPCL model in (**B**) 24- and (**C**) 96-well plates. The relative contraction area was calculated as a percentage of the Day 1 comPDAC-FPCL area. Results are presented as mean ± standard deviation (SD; n = 4). H&E staining of comPDAC-FPCL models in (**D**) 24- and (**E**) 96-well plates (Days 7 and 14). Scale bar: 200 μm.

**Figure 2 ijms-25-03740-f002:**
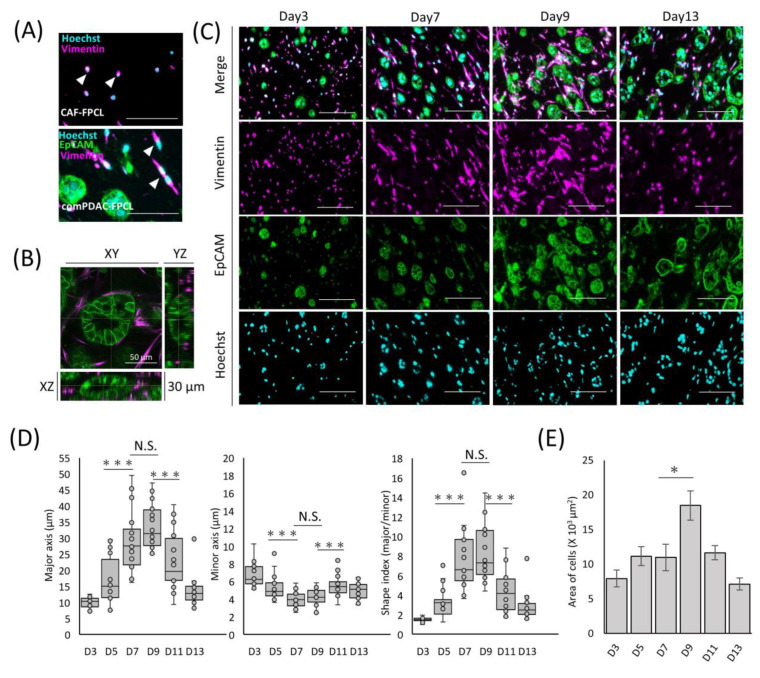
Differential morphological dynamics of CAFs within comPDAC-FPCL models. (**A**) Morphological comparison of CAF progenitors in the CAF-FPCL and comPDAC-FPCL models (Day 7). Capan-1 cells stained for epithelial cell adhesion molecule (EpCAM; Green), CAF progenitors stained for vimentin (Magenta), and nuclear stain for Hoechst (Cyan). Arrowheads indicate CAF progenitor morphologies. Scale Bar: 50 μm. (**B**) 3D imaging of the glandular structure of CAF-surrounded PDAC. Z: 30 μm. Scale Bar: 50 μm. (**C**) Dynamic changes in Capan-1 cell and CAF progenitor morphologies within the comPDAC-FPCL models. Scale Bar: 100 μm. (**D**) Dynamics of the quantitative value of CAF progenitors in the comPDAC-FPCL models. Shape Index indicates the ratio between the major and minor axis lengths. Results are presented as mean ± SD (n = 20). * *p* < 0.05, *** *p* < 0.001, N.S., not significant, unpaired Student’s *t*-test. (**E**) Dynamics of vimentin-positive cells in the mean cellular area. Results are presented as mean ± SD (n = 3). * *p* < 0.05, unpaired Student’s *t*-test.

**Figure 3 ijms-25-03740-f003:**
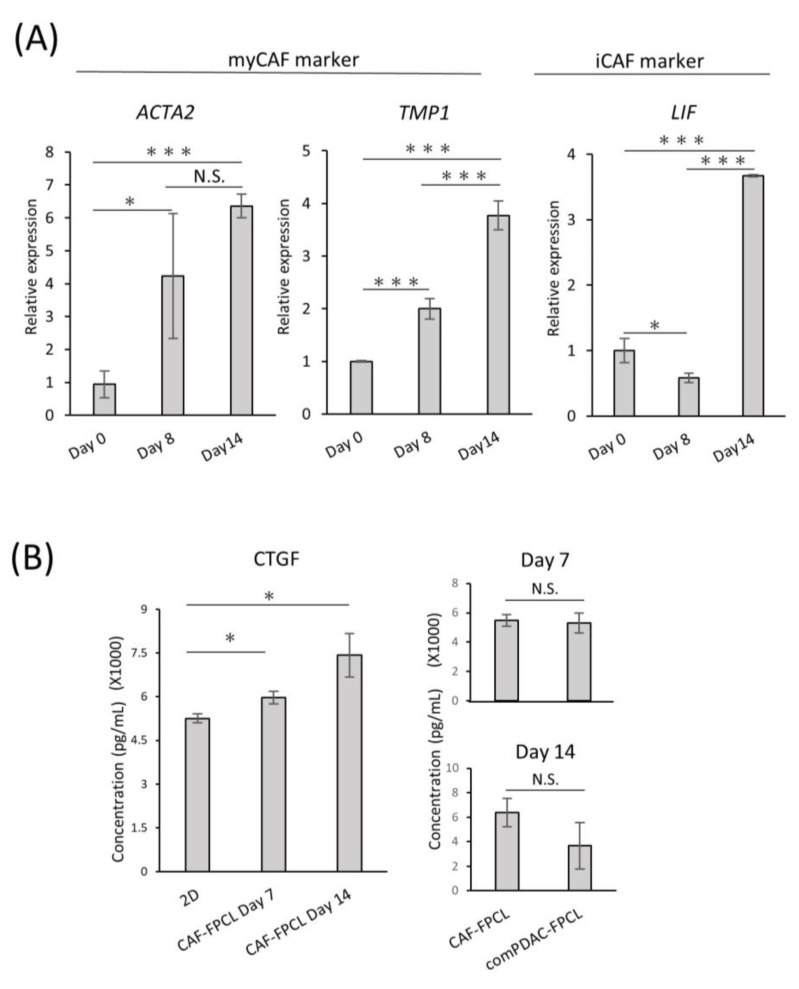
Differential expression and temporal dynamics of CAF subtypes. (**A**) Quantitative real-time PCR (qPCR) analysis of myCAF and iCAF markers on Days 0, 8, and 14 of the comPDAC-FPCL models. Day 0 is after 3 h of modeling. Results are presented as mean ± SD (n = 3). * *p* < 0.05, *** *p* < 0.001, N.S., not significant, unpaired Student’s *t*-test. (**B**) Enzyme-linked immunosorbent assay analysis of CTGF in the conditioned medium of 2D CAF progenitors, CAF-FPCL, and comPDAC-FPCL. Results are presented as mean ± SD (n = 3). * *p* < 0.05, N.S., not significant, unpaired Student’s *t*-test.

**Figure 4 ijms-25-03740-f004:**
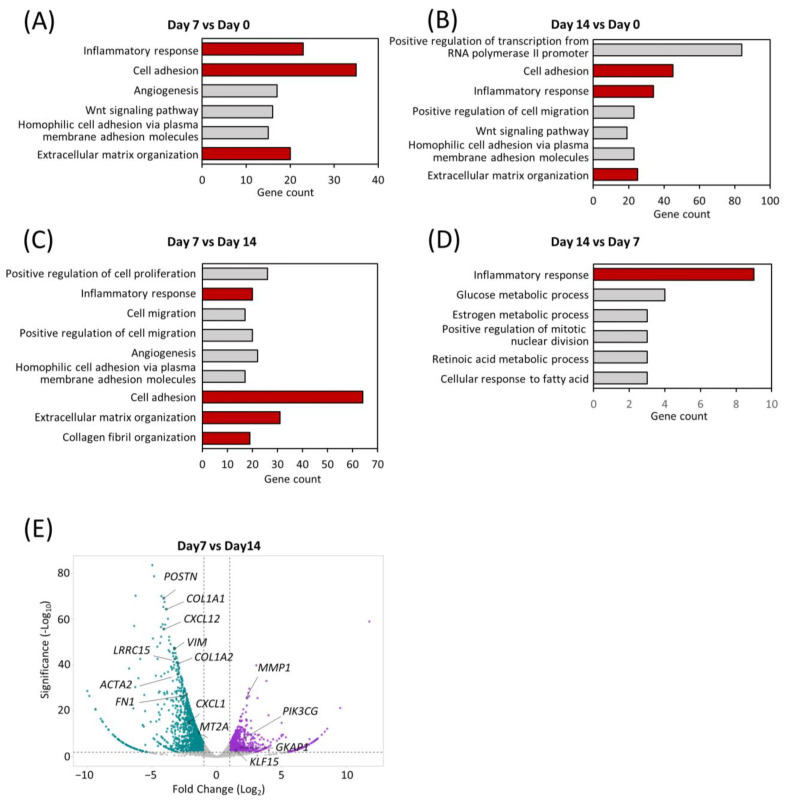
Transcriptomic dynamics reveal structural and metabolic alterations. GO enrichment analysis of differentially expressed genes at (**A**) early (Day 7) and (**B**) late (Day 14) stages in comPDAC-FPCL models compared to the initial models (Day 0). CAF-related GO terms are shown in red. GO enrichment analysis of differentially expressed genes between (**C**) early (Day 7) and (**D**) late (Day 14) stages in comPDAC-FPCL models. Tumor-active microenvironment-related GO terms are shown in red; log_2_ [fold change] ≥ 2. Volcano plot showing differentially expressed genes between (**E**) early (Day 7) and late (Day 14) stages in comPDAC-FPCL models (adjusted *p*-value < 0.05 and log_2_ [fold change] ≥ 1). The upregulated genes in comPDAC-FPCL models (Day 7) and comPDAC-FPCL models (Day 14) are shown in cyan and magenta, respectively.

**Figure 5 ijms-25-03740-f005:**
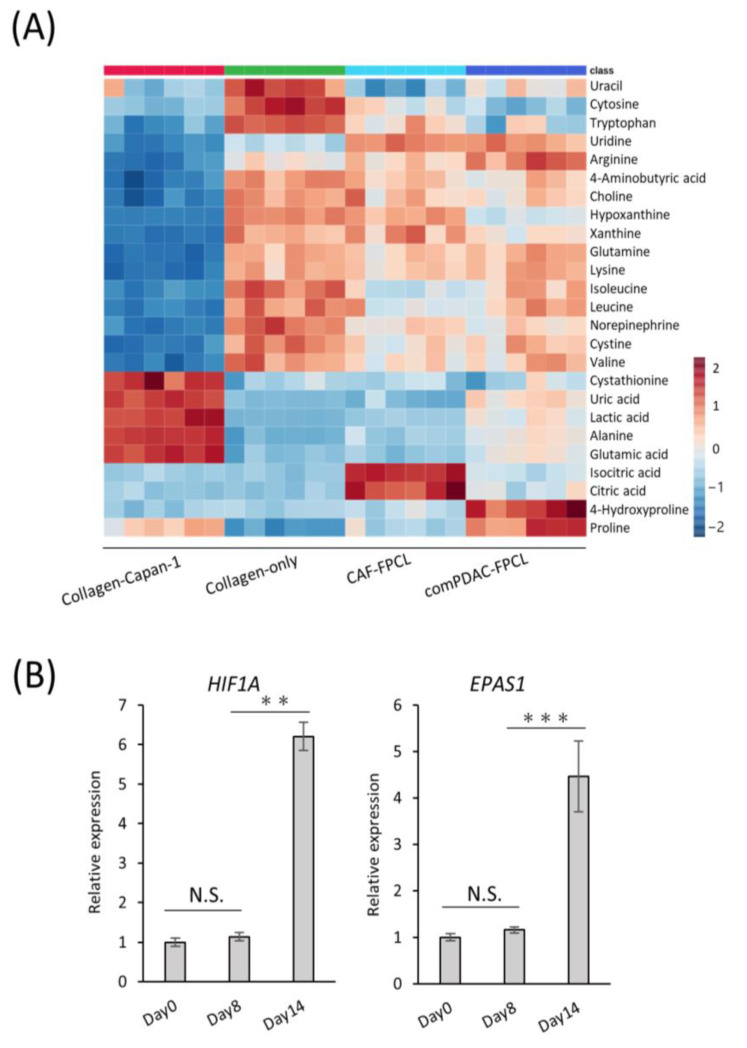
Characterization of extracellular metabolite dynamics. (**A**) Heatmap comparing altered metabolites between supernatant from Collagen-Capan-1, Collagen-only, CAF-FPCL, and comPDAC-FPCL (n = 6 each). (**B**) qPCR analysis of hypoxia-related markers HIF1A and EPAS1 on Days 0, 8, and 14 in comPDAC-FPCL models. Results are presented as mean ± SD (n = 3). ** *p* < 0.01, *** *p* < 0.001, N.S., not significant, unpaired Student’s *t*-test.

**Figure 6 ijms-25-03740-f006:**
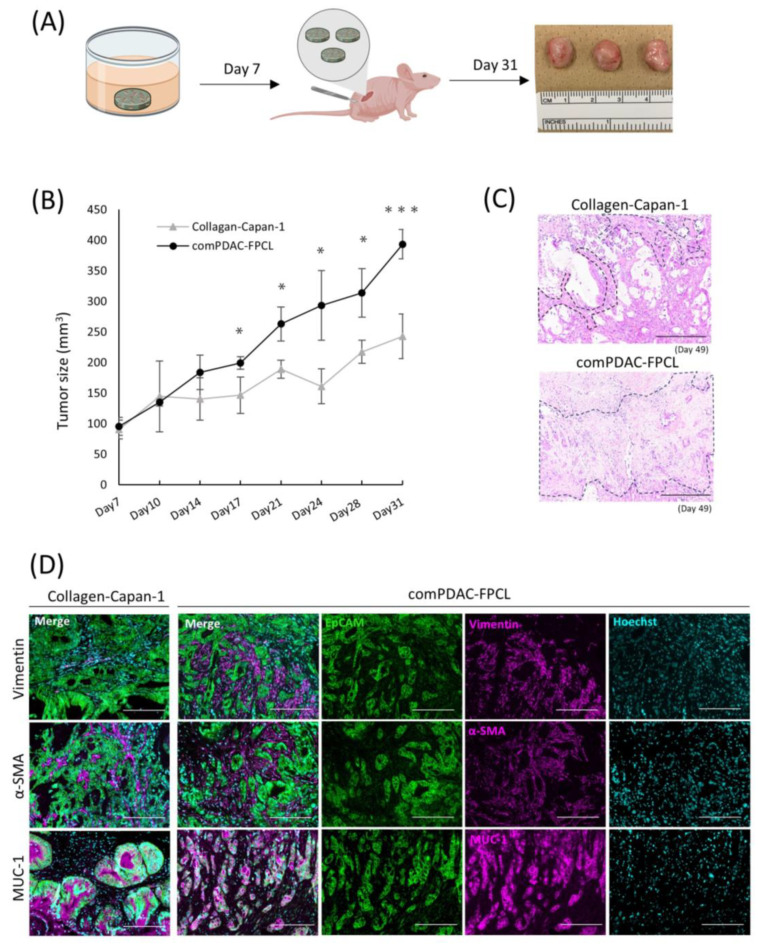
Validation of the tumorigenic potential of 3D tissue generated by the comPDAC-FPCL models in vivo. (**A**) Schematic diagram of the in vivo experimental workflow. (**B**) Tumor growth curve on Day 31. Results are presented as mean ± SD (n = 3). * *p* < 0.05, *** *p* < 0.001, unpaired Student’s *t*-test. (**C**) Histological analysis of H&E-stained tumor sections generated by Collagen-Capan-1 and comPDAC-FPCL. The stromal area is delineated by the black dashed line. Scale Bar: 200 μm. (**D**) Representative immunofluorescence image of Collagen-Capan-1- and comPDAC-FPCL-generated tumors. Capan-1 cells stained for EpCAM (Green) and MUC-1 (Magenta). CAF progenitors stained for vimentin/α-SMA (Magenta). All nuclear stains for Hoechst (Cyan). Scale Bar: 200 μm.

**Figure 7 ijms-25-03740-f007:**
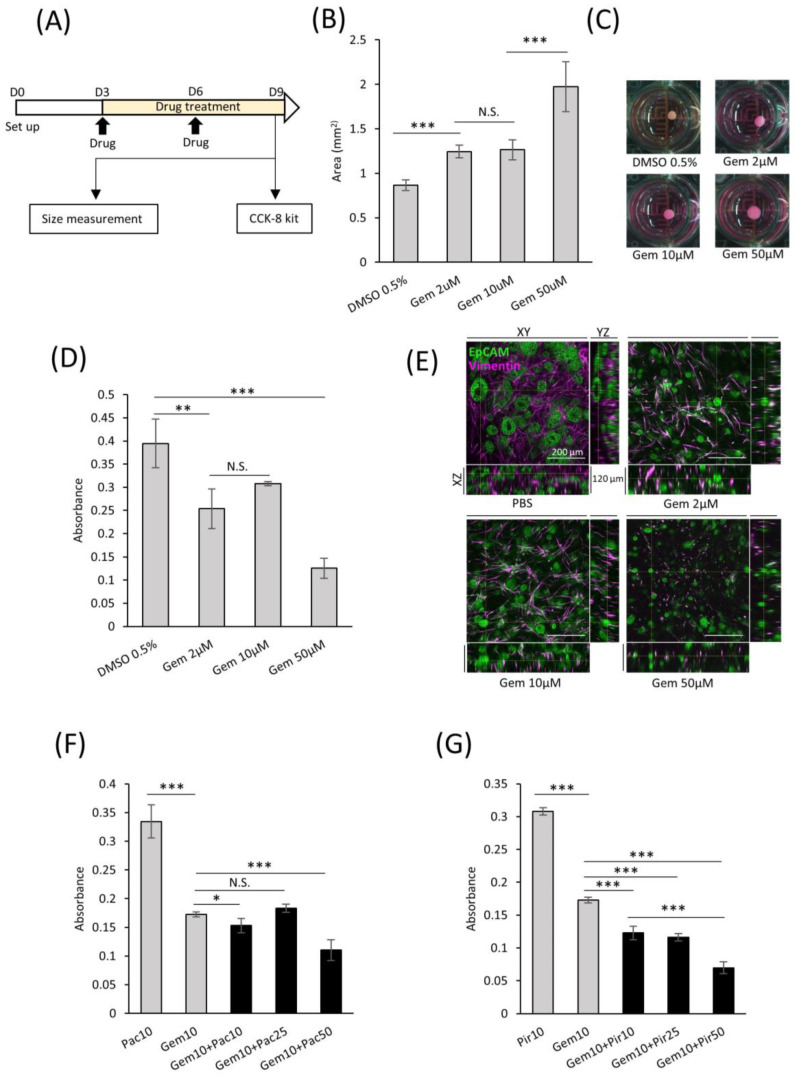
Applicability of comPDAC-FPCL models for drug screening. (**A**) Schematic diagram of the protocol for drug screening using the comPDAC-FPCL models. (**B**) Single-drug screening of comPDAC-FPCL models estimated using size changes. Results are presented as mean ± SD (n = 4). *** *p* < 0.001, N.S., not significant, unpaired Student’s *t*-test. (**C**) Images of single-drug screening. (**D**) Single-drug screening of comPDAC-FPCL models estimated using a Cell Counting Kit-8 (CCK-8) assay. Results are presented as mean ± SD (n = 4). *** *p* < 0.001, ** *p* < 0.01, N.S., not significant, unpaired Student’s *t*-test. (**E**) 3D imaging within the comPDAC-FPCL models after Gem treatment. Z stack: 120 μm. Scale Bar: 200 μm. (**F**,**G**) Multi-drug screening of comPDAC-FPCL models estimated using a CCK-8 assay. Multi-drug treatment of (**F**) Gem+Pac and (**G**) Gem+Pir. Results are presented as mean ± SD (n = 4). * *p* < 0.05, *** *p* < 0.001, N.S., not significant, unpaired Student’s *t*-test. Gem, gemcitabine. Pac, paclitaxel. Pir, pirfenidone.

## Data Availability

All other data supporting the findings of this study are available from the corresponding author upon request.

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
