# Peer review of "Collagen Lattice Model, Populated with Heterogeneous Cancer-Associated Fibroblasts, Facilitates Advanced Reconstruction of Pancreatic Cancer Microenvironment"

_ijms, 2024, doi:10.3390/ijms25073740_

Round 1

Reviewer 1 Report

Comments and Suggestions for Authors Song et al. present a collagen lattice model to reconstruct pancreatic cancer microenvironment. Recently, the relevance of the tumor microenvironment including PDAC is crucial. Specifically  in the contexts of drug efficacies, a true 3-D model is highly significant. There are some suggestions that can help to enhance the impact of this paper. 1. The authors are encouraged to include a section on tumor microenvironment and problems such as drug resistance.  2. However, the authors have profiled the metabolites specifically amino acids. It would be pertinent to analyze at the intracellular and extracellular levels. 3. Also, authors should quantify the levels of tested anticancer drugs accumulated at the intracellular levels.  4. The authors may discuss the extension of the proposed 3-D model into the futuristic 4-D PDAC model that can mimic microenvironment even better than the proposed model. 5. The authors should emphasize on the relevance of the proposed model on preclinical and clinical drug discovery for PDAC and other tumor types. 6. Using the PDAC-FPCL model, the authors have tested anticancer drugs such as gems and PAC. What was the IC50 values? Data do not show clearly the increase in drug concentration and proportionate cell toxicity. Also, whether authors compared with well known cytotoxicity models.  Comments on the Quality of English Language

Moderate.

Reviewer 2 Report

Comments and Suggestions for Authors

The manuscript by Xiaoyu Song explores the establishment of a 3D collagen 1 matrix model to co-cultivate pancreatic cancer cells and fibroblasts. The authors investigate molecular shifts in cancer-associated fibroblast (CAF) polarization from a contractile subtype to a more inflammatory one within this model. They also monitor the growth of cancer cells embedded in collagen matrices with the addition of CAFs both in vitro and in vivo. Furthermore, the sensitivity of pancreatic ductal adenocarcinoma (PDAC) cells to gemcitabine is assessed in vitro within this model. The study is well-written and presents intriguing findings. However, several limitations within the manuscript need to be addressed before publication.

1.      It is known from previous studies that collagen 1 dimers produced by CAFs can restrict tumor growth in PDAC models (please refer to Chen, Y.; Cancer Cell 2022, 40, 818–834.e9. doi: 10.1016/j.ccell.2021.02.007; Tian, C.; Nat. Commun. 2021, 12, 2328. DOI: 10.1038/s41467-021-22490-9). However, this aspect has not been reflected in the current study, leading to confusion. To address this potential issue with using collagen 1 matrix, the authors should conduct a straightforward experiment to assess the impact of collagen 1 on tumor growth. In this experiment with 3D collagen beds, an important control would involve monitoring the growth of 1. CAPAN cells without any matrix; 2. CAPAN cells with collagen; 3. CAPAN cells with CAFs without collagen, and 4. CAPAN cells with both CAFs and collagen. Tumor cell growth could be monitored by incorporating luciferase or employing another method sufficient to separate CAFs and CAPAN cells.

2.      Ideally, this experiment should also be conducted in vivo, although I understand that it might be challenging.

3.      The dual role of collagen 1 dimers should be discussed in the manuscript.

4.      RNAseq data on the induction of inflammatory cancer-associated fibroblast (iCAF) gene expression by day 14 should be manually validated by qPCR. For example, more inflammatory genes besides LIF could be included in Fig. 3A. Only one gene, LIF, is not sufficient for comprehensive validation.

Reviewer 3 Report

Comments and Suggestions for Authors

Pancreatic cancer is a very aggressive and common carcinoma with the dominant occurrence of pancreatic ductal adenocarcinoma (PDAC). Typically, there is a low correlation between initial in vitro activity of potential drugs and in vivo and clinical efficacy. Song et al. developed a novel 3D PDAC model exhibiting clinically similar glandular structures surrounded by spindle-shaped stromal cells, which better simulate real tumor biological and physical properties.

In my opinion, the research is well-designed. PDAC contains a substantial proportion of heterogeneous cancer-associated fibroblasts (CAFs). Therefore, to create a 3D model of PDAC, the fibroblast-populated collagen lattice (FPCL) model was used, called the combined PDAC-FPCL model (omPDAC-FPCL), where PDAC cells (Capan-1) and CAFs coexist. Structural and functional analyses of this model (especially on the 7th and 14th day) confirmed its usefulness in screening for improved therapy. These models mirror the hypoxic conditions and mechanical stress inherent in the PDAC microenvironment, and the proposed protocol for screening potential drug candidates using comPDAC-FPCL models is worth the attention of researchers working in the field of PDAC.

I found two minor issues which might be corrected/explained:

1. Figure 2D—I think there is a lack of explanation of which kind of model is presented in the two first plots (Day 7? and Day 14?).

2. Lines 278-279 
"(Figure 7D). Expectedly, they showed therapeutic advantages at 50, 10, and 2 278 μM, whereas 10 and 2 μM yielded slight therapeutic outcomes."
I do not understand this interpretation of results in comparison to Figure 7D. Can you check and explain or correct it?

Round 2

Reviewer 2 Report

Comments and Suggestions for Authors

The revised manuscript has expanded the discussion section to include more information on the role of collagen in tumor cell physiology and drug accessibility. However, the authors failed to mention that heterotrimeric collagen 1, specifically produced by fibroblasts, restricts tumor growth, suggesting that the presence of normal collagen may actually help suppress pancreatic cancer growth. In light of this, the authors should include proper controls in their study, particularly two groups of samples: tumor cells alone and tumor cells with collagen, to assess the growth of tumor cells. Without these controls, the study lacks consistency and is not convincing in this regard. The reference to the previous study by the group (https://doi.org/10.3390/bioengineering10121437) also does not demonstrate the direct effect of collagen 1 on tumor cell proliferation. Therefore, I strongly recommend including this control in the study.

Furthermore, I would like to reiterate that the authors' conclusion regarding the shift to inflammatory cancer-associated fibroblasts (CAFs) by day 14 is not supported by the data. The mRNA level of LIF alone is not sufficient evidence. Figure 4 demonstrates different gene signatures of fibroblasts, indicating that the authors obtained a full list of markers. They could then sort out inflammatory genes and include a few of them in Figure 3A. Additionally, please discuss whether induction of hypoxia by day 14 could be a driver of increased ongoing inflammation.

Round 3

Reviewer 2 Report

Comments and Suggestions for Authors

The manuscript has been improved by discussing more intensively the role of collagen dimers. In its present form, the manuscript provides valuable insights into the model with tumor cells and fibroblasts embedded in a collagen matrix. This model is primarily useful for evaluating the impact of the collagen matrix on drug sensitivity. However, it is not suitable for assessing the proliferation of tumor cells in the conditions of a collagen matrix because this has not been demonstrated. Additionally, supportive features of cancer-associated fibroblasts (CAFs) might not be dependent on collagen, as this has not been demonstrated either.

The GO and GSEA analysis published in Song et al. (Bioengineering, 2023) still do not conclusively prove that collagen enhances the growth of CAPAN cells. This analysis suggests that differently expressed genes are associated with proliferation, but the nature of this association should be validated experimentally. Moreover, the high glycolysis observed may be linked not to high proliferation, but to high hypoxia induced by cultivation in collagen, and does not necessarily indicate increased dividing capacity of tumor cells.

I recommend including one more inflammatory marker into Figure 3A to provide a more comprehensive representation of inflammatory gene expression.

Since the manuscript still holds value for researchers seeking models to monitor drug sensitivity in the conditions of collagen and hypoxia, as well as for studying contractility and CAF physiology in complex systems, it can be published. However, for future studies, the authors must include a proper control with only CAPAN cells and CAPAN cells in a collagen matrix.

Author Response

Thank you very much for your valuable comments and insights. We greatly appreciate the considerable effort you have put into reviewing our manuscript and include many important points that we intend to utilize in our future research. We are deeply grateful for the time and effort you have dedicated to providing us with this constructive feedback. The insights provided will significantly contribute to the depth of our research and help us achieve better outcomes in the future.